# Validation of Space-Based Albedo Products from Upscaled Tower-Based Measurements Over Heterogeneous and Homogeneous Landscapes

**Rui Song [1,*], Jan-Peter Muller [1], Said Kharbouche [1], Feng Yin [2], William Woodgate [3], Mark Kitchen [3], Marilyn Roland [4], Nicola Arriga [4,5], Wayne Meyer [6], Georgia Koerber [6], Damien Bonal [7], Benoit Burban [8], Alexander Knohl [9], Lukas Siebicke [9], Pauline Buysse [10], Benjamin Loubet [10], Montagnani Leonardo [11,12], Christophe Lerebourg [13] and Nadine Gobron [5]**

[1] Imaging Group, Mullard Space Science Laboratory, Department of Space & Climate Physics, University College London, Holmbury St Mary, Surrey RH56NT, UK; j.muller@ucl.ac.uk (J.-P.M.); s.kharbouche@ucl.ac.uk (S.K.)

[2] NCEO, Department of Geography, University College London, Gower Street, London WC1E 6BT, UK; feng.yin.15@ucl.ac.uk

[3] CSIRO, Black Mountain, Building 801, Canberra 2601, Australia; William.Woodgate@csiro.au (W.W.); Mark.Kitchen@csiro.au (M.K.)

[4] Research Center of Excellence PLECO, Department of Biology, University of Antwerp, Universiteitsplein 1, B-2610 Wilrijk, Belgium; marilyn.roland@uantwerpen.be

[5] Joint Research Centre, European Commission, Via Enrico Fermi 2749, 21027 Ispra, Italy; Nicola.ARRIGA@ec.europa.eu (N.A.); nadine.gobron@ec.europa.eu (N.G.)

[6] TerRésultats de recherche TERN - Terrestrial Ecosystem Research Network (TERN), School of Biological Sciences, The University of Adelaide, Adelaide 5005, Australia; Wayne.meyer@adelaide.edu.au (W.M.); georgia.koerber@adelaide.edu.au (G.K.)

[7] AgroParisTech, INRAE, UMR Silva, Université de Lorraine, 54000 Nancy, France; damien.bonal@inra.fr

[8] AgroParisTech, INRAE, UMR EcoFoG, Cirad, CNRS, Université des Antilles, Université de Guyane, 97310 Kourou, France; benoit.burban@ecofog.gf

[9] Bioclimatology, Faculty of Forest Sciences and Forest Ecology, University of Goettingen, 37077 Goettingen, Germany; aknohl@uni-goettingen.de (A.K.); Lukas.Siebicke@forst.uni-goettingen.de (L.S.)

[10] Institut National de Recherche Agronomique (INRA), Université Paris-Saclay, 78850 Saint-Aubin, France; pauline.buysse@inra.fr (P.B.); Benjamin.Loubet@inra.fr (B.L.)

[11] Faculty of Science and Technology, Free University of Bozen-Bolzano, 39100 Bolzano, Italy

[12] Forest Services, Autonomous Province of Bolzano, 39100 Bolzano, Italy; leonardo.montagnani@unibz.it

[13] ACRI-ST, 260 Route de Pin Montard, BP 234, 06904 Sophia Antipolis, France; christophe.lerebourg@acri-st.fr

\* Correspondence: rui.song@ucl.ac.uk

**Abstract:** Surface albedo is a fundamental radiative parameter as it controls the Earth's energy budget and directly affects the Earth's climate. Satellite observations have long been used to capture the temporal and spatial variations of surface albedo because of their continuous global coverage. However, space-based albedo products are often affected by errors in the atmospheric correction, multi-angular bi-directional reflectance distribution function (BRDF) modelling, as well as spectral conversions. To validate space-based albedo products, an in situ tower albedometer is often used to provide continuous "ground truth" measurements of surface albedo over an extended area. Since space-based albedo and tower-measured albedo are produced at different spatial scales, the can be directly compared only for specific homogeneous land surfaces. However, most land surfaces are inherently heterogeneous with surface properties that vary over a wide range of spatial scales. In this work, tower-measured albedo products, including both directional hemispherical reflectance (DHR) and bi-hemispherical reflectance (BHR), are upscaled to coarse satellite spatial resolutions using a new method. This strategy uses high-resolution satellite derived surface albedos to fill the gaps between the albedometer's field-of-view (FoV) and coarse satellite scales. The high-resolution

surface albedo is generated from a combination of surface reflectance retrieved from high-resolution Earth Observation (HR-EO) data and moderate resolution imaging spectroradiometer (MODIS) BRDF climatology over a larger area. We implemented a recently developed atmospheric correction method, the Sensor Invariant Atmospheric Correction (SIAC), to retrieve surface reflectance from HR-EO (e.g., Sentinel-2 and Landsat-8) top-of-atmosphere (TOA) reflectance measurements. This SIAC processing provides an estimated uncertainty for the retrieved surface spectral reflectance at the HR-EO pixel level and shows excellent agreement with the standard Landsat 8 Surface Reflectance Code (LaSRC) in retrieving Landsat-8 surface reflectance. Atmospheric correction of Sentinel-2 data is vastly improved by SIAC when compared against the use of in situ AErosol RObotic NETwork (AERONET) data. Based on this, we can trace the uncertainty of tower-measured albedo during its propagation through high-resolution EO measurements up to coarse satellite scales. These upscaled albedo products can then be compared with space-based albedo products over heterogeneous land surfaces. In this study, both tower-measured albedo and upscaled albedo products are examined at Ground Based Observation for Validation (GbOV) stations (https://land.copernicus.eu/global/gbov/), and used to compare with satellite observations, including Copernicus Global Land Service (CGLS) based on ProbaV and VEGETATION 2 data, MODIS and multi-angle imaging spectroradiometer (MISR).

**Keywords:** surface albedo; directional hemispherical reflectance; bi-hemispherical reflectance; upscaling; CGLS; ProbaV; vegetation; MODIS; MISR

## 1. Introduction

Surface albedo, or the integrated hemispherical surface reflectance, is the ratio of the radiant flux reflected from the Earth's surface to the incident radiant flux. Surface albedo is a fundamental radiative parameter as it controls the Earth's energy budget by determining the amount of solar radiation absorbed by the surface [1]. Since surface albedo is highly variable in space and time over natural landscapes, it is necessary to have a long-term record of accurately estimated surface albedo at appropriate spatial scales. It is also a key parameter to assess the impacts of surface warming, especially in polar regions, such as Greenland [2].

Satellite observations provide a unique tool for monitoring surface albedo on a global scale. To retrieve surface albedo from satellite-based instruments, sufficient numbers of directional surface reflectance measurements are needed in order to model the bidirectional reflectance distribution function (BRDF). Such directional measurements of surface reflectance can be obtained from a single field of view sensor by accumulating sequential measurements over a period of time (e.g., moderate resolution imaging spectroradiometer (MODIS) over 16 days), or from multi-angular sensors by directly obtaining directional measurements near simultaneously (e.g., multi-angle imaging spectroradiometer (MISR) within 7 minutes). Over the last two decades, global land surface albedo products have been generated from various satellite-based instruments, including MODIS [3], MISR [4], Advanced Very High Resolution Radiometer (AVHRR) [5], and more [6,7]. Surface albedo retrieved from satellite observations are frequently contaminated by noise from atmospheric corrections to convert top-of-atmosphere (TOA) reflectances to surface reflectances, using narrow-to-broadband conversions to transform spectral albedo to broadband albedo, as well as using insufficient multi-angular measurements in BRDF modelling. In addition, there are many instances of residual cloud contamination effects [8]. In situ tower albedometers, which can provide "ground truth" of surface albedo over a field-of-view (FoV) from tens to a few hundred metres, are therefore often used to validate satellite retrieved coarse-resolution albedo products. Typical albedometers consist of two calibrated pyranometers, with one measuring downwelling solar radiation and the other measuring surface reflected upwelling radiation. The Baseline Surface Radiation Network (BSRN) was designated as the global baseline network for surface radiation for the World Meteorological Organization (WMO) and World Climate

Research Programme (WCRP), and started providing systematic albedometer observations since 1992 from 10-m high towers [9]. The surface radiation budget network (SURFRAD), which is also a contributor to the BSRN, has maintained an excellent record of solar radiation at tower stations in the US since 1995 [10]. FLUXNET [11] is also a global network of micrometeorological tower sites, but focused mainly on measurements of carbon dioxide and water fluxes. Many of these FLUXNET sites include tower-based albedometers covering a wide range of different land cover types including forests, croplands, grasslands, wetlands, etc. (fluxnet.ornl.gov).

A number of studies have attempted to validate satellite-based albedo products by directly comparing them for specific spatially representative tower albedometer measurement sites [12,13]. These studies assume that the tower-measured albedo over a limited footprint (tens to a few hundred metres) can represent the satellite-measured albedo at coarse resolutions (500 m–3 km), so that no upscaling is needed. They often employ spatial autocorrelation with high resolution EO images to assess the degree of homogeneity. However, most of the Earth's land surface is not homogeneous in surface reflectivity. This direct "point-to-pixel" scaling is not always applicable, as albedo measured at tower albedometer FoV can occasionally have stronger correlations with adjacent satellite pixels over the closest pixel if we have two strongly contrasting surface albedos, due to having cropland adjoining forest (Said Kharbouche, private communication, 2017). To overcome this limitation, a method is needed which is able to upscale tower-measured albedo to satellite spatial resolutions over both homogeneous and heterogeneous sites. Song et al. [14] developed a framework for upscaling surface albedo from ground level to coarse satellite scales using a combination of downscaled MODIS BRDF climatology and high-resolution EO. This method has two major components, one deriving high-resolution albedo products from the MODIS/high-resolution Earth Observation (HR-EO) combination, and the other using derived high-resolution albedo to fill the gaps between tower FoV and coarse satellite scales. This method makes it possible to validate satellite albedo products using upscaled tower-measured albedo over both homogeneous and heterogeneous sites, and has been demonstrated over 20 Ground Based Observation for Validation (GbOV) tower sites covering different landscapes. In this work, we have further developed this processing method by employing a novel Sensor Invariant Atmospheric Correction (SIAC) approach to obtain high-resolution surface reflectance/albedo more accurately including explicitly the effects of surface BRDF, and providing a traceable uncertainty estimate for the upscaled albedo values. The SIAC atmospheric correction method developed by Yin et al. [15] can accurately retrieve satellite-based spectral surface reflectance, along with deriving an uncertainty estimate for every single pixel. This allows the uncertainty of tower-measured albedo to be traceable during its propagation through high-resolution EO measurements up to coarse satellite scales. In this paper, we aim to demonstrate that:

(1) The SIAC method has a good agreement with the Landsat 8 Surface Reflectance Code (LaSRC) algorithm in retrieving Landsat-8 surface reflectance. We also show that the SIAC method has better performance than the Sen2cor tool in retrieving Sentinel-2 surface reflectance.

(2) a streamlined version of SIAC, which includes a representation of anisotropy or surface directional/structural/topography dependence into the upscaling framework, improves the accuracy of upscaled tower albedo values.

(3) The upscaled albedo products, including direct hemispherical reflectance (DHR) and bi-hemispherical reflectance (BHR) for three different global networks are examined for the first time over heterogeneous sites selected from the GbOV tower stations in addition to homogeneous sites.

## 2. Materials and Methods

### 2.1. Ground Measurements

In [14], Song et al. compared space-based albedo measurements against tower measurements between the year 2012 and 2016 at 20 GbOV tower sites selected from the FLUXNET, SURFRAD

and BSRN networks. In this study, all the available tower measurements are extended to cover the years 2017 and 2018, except for the DE-GEB, US-BAO, US-BRW and SPO sites, which have been replaced by the NL-CAB, NM-GOB, NO-NYA and DOM sites. The original site list for 2012-2016 can be found in [14]. The new sites are now located over 6 continents as follows: Europe, North America, South America, Africa, Australia, and Antarctic, as shown in Figure 1. Key characteristics of the new 20 sites are listed in Table 1, including geographic coordinates, associated networks, footprint area, land cover types, and time range of the processed tower albedo data. Figure 2 shows an example of the 42m tower at the Hainich FLUXNET tower station (DE-HAI), and configuration of the albedometer. The albedometer is located at the extreme right with one part facing upwards and the other part facing downwards. Figure 3 also shows a pyranometer for measuring diffuse solar radiation.

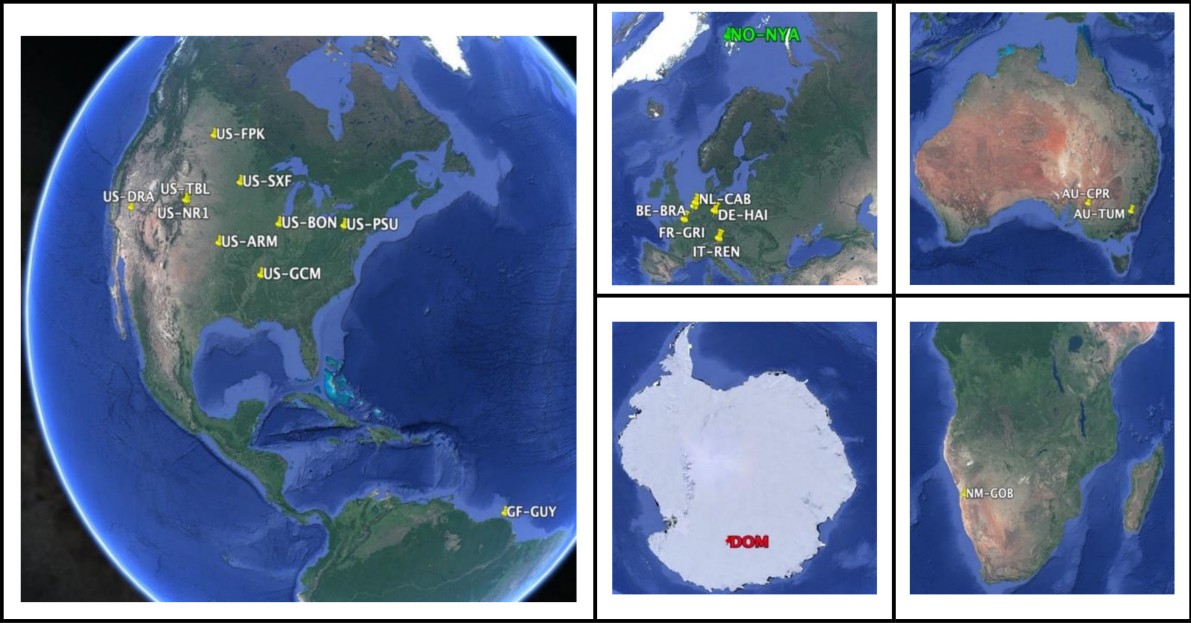

**Figure 1.** Geographical distribution of selected sites (Google Earth).

**Table 1.** List of tower sites with key characteristics: acronyms, geographical coordinates, networks, footprints and land cover types defined by International Geosphere-Biosphere Programme (IGBP).

| Station | Acronym | Latitude (°) | Longitude (°) | Network | Footprint | Land Classification (IGBP) | Time Range |
|---|---|---|---|---|---|---|---|
| Ny-Ålesund** | NO-NYA | 78.925 | 11.93 | BSRN (http://bsrn.awi.de) | 25 m | Snow and ice | 2017–2018 |
| Concordia Station* | DOM | -75.1 | 123.383 | BSRN | 46 m | Snow and Ice | 2017–2018 |
| Cabauw | NL-CAB | 51.971 | 4.927 | BSRN | 46 m | Grasslands | 2016–2018 |
| Gobabeb* | NM-GOB | -23.519 | 15.083 | BSRN | 46 m | Desert | 2016–2018 |
| Niwot Ridge# | US-NR1 | 40.033 | −105.546 | FLUXNET (https://FLUXNET.ornl.gov) | 158 m | Evergreen Needleleaf | 2013–2018 |
| ARM Southern Great Plains | US-ARM | 36.606 | −97.489 | FLUXNET | 25 m | Croplands | 2012–2018 |
| Hainich* | DE-HAI | 51.070 | 10.450 | FLUXNET | 265 m | Mixed Forest | 2012–2018 |
| Grignon | FR-GRI | 48.844 | 1.952 | FLUXNET | 67 m | Croplands | 2012–2018 |
| Guyaflux*# | GF-GUY | 5.279 | −52.925 | FLUXNET | 290 m | Evergreen Broadleaf | 2012–2018 |
| Brasschaat# | BE-BRA | 51.309 | 4.521 | FLUXNET | 240 m | Mixed Forest | 2012–2018 |
| Renon | IT-REN | 46.587 | 11.434 | FLUXNET | 152 m | Evergreen Needleleaf | 2012–2017 |
| Tumbarumba* | AU-TUM | −35.657 | 148.152 | FLUXNET | 505 m | Evergreen Broadleaf | 2012–2018 |
| Calperum # | AU-CPR | −34.003 | 140.588 | FLUXNET | 215 m | Closed Shrublands | 2013–2018 |
| Sioux Falls | US-SXF | 43.730 | −96.620 | SURFRAD (https://www.esrl.noaa.gov/gmd/grad/surfrad/) | 126 m | Croplands | 2012–2018 |
| Bondville | US-BON | 40.052 | −88.373 | SURFRAD | 126 m | Croplands | 2012–2018 |
| Desert Rock * | US-DRA | 36.624 | −116.019 | SURFRAD | 126 m | Open Shrublands | 2012–2018 |
| Fort Peck * | US-FPK | 48.308 | −105.102 | SURFRAD | 126 m | Grasslands | 2012–2018 |
| Goodwin Creek | US-GCM | 34.255 | −89.873 | SURFRAD | 126 m | Deciduous Broadleaf | 2012–2018 |
| Penn State | US-PSU | 40.720 | −77.931 | SURFRAD | 126 m | Deciduous Broadleaf | 2012–2018 |
| Table Mountain * | US-TBL | 40.125 | −105.237 | SURFRAD | 126 m | Bare soil and Rocks | 2012–2018 |

Sites marked with * have been claimed to be spatially representative, which is sometimes referred to as homogeneous by [16]. ** NO-NYA is spatially representative during snow covered periods, but heterogeneous during the snow melt season. N.B. Sites marked with # do not have diffuse radiation measurements that cover the entire time period, so a method to model this taken from [17] is used to estimate diffuse radiation. Further details of albedometers can be found in the Algorithm Theoretical Basis Documents (ATBD) (https://gbov.acri.fr/public/docs/products/2019-11/GBOV-ATBD-RM1-LP1-LP2_v1.3-Energy.pdf).

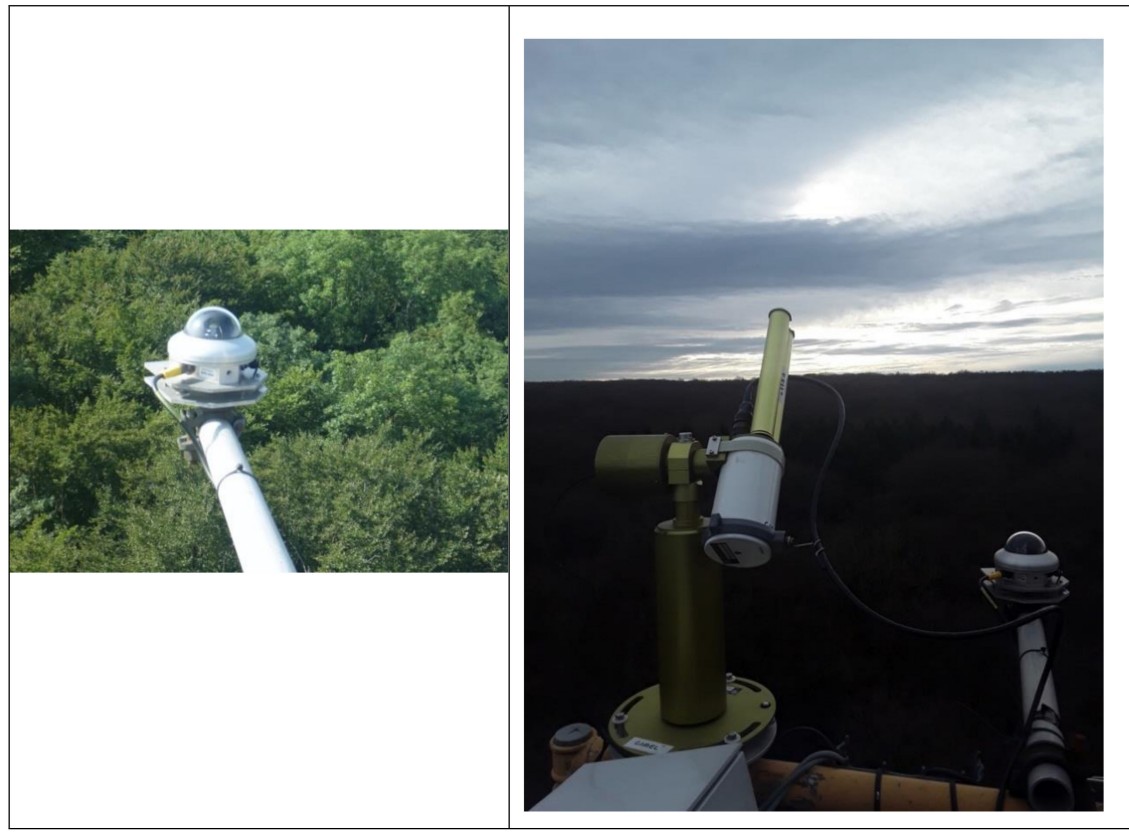

**Figure 2.** Photos taken at the Hainich FLUXNET tower site, and configuration of the albedometer (and AErosol RObotic NETwork (AERONET) CIMEL sun photometer. Courtesy of Frank Tindemann (left) and Lukas Siebicke (right), Universität Göttingen.

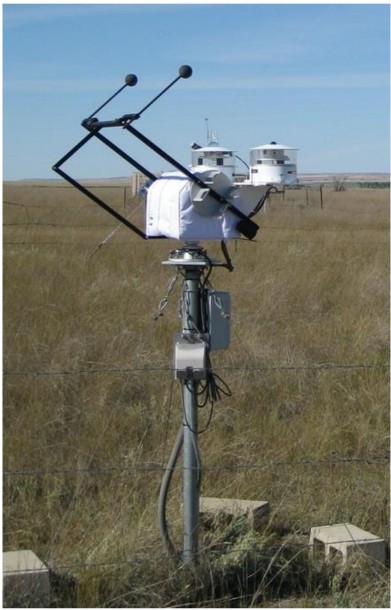

**Figure 3.** Example of pyranometer for measuring diffuse radiation. An accurate sun-tracker is installed to shield the pyranometer from direct solar beam. Photo is courtesy of surface radiation budget network (SURFRAD) US-FPK site.

Figure 4 shows the time series of raw measurements for the newly added NL-CAB site. It includes three key variables: incoming shortwave radiation, outgoing shortwave radiation and diffuse shortwave radiation, which are essential for the BRDF angular modelling and surface albedo calculations.

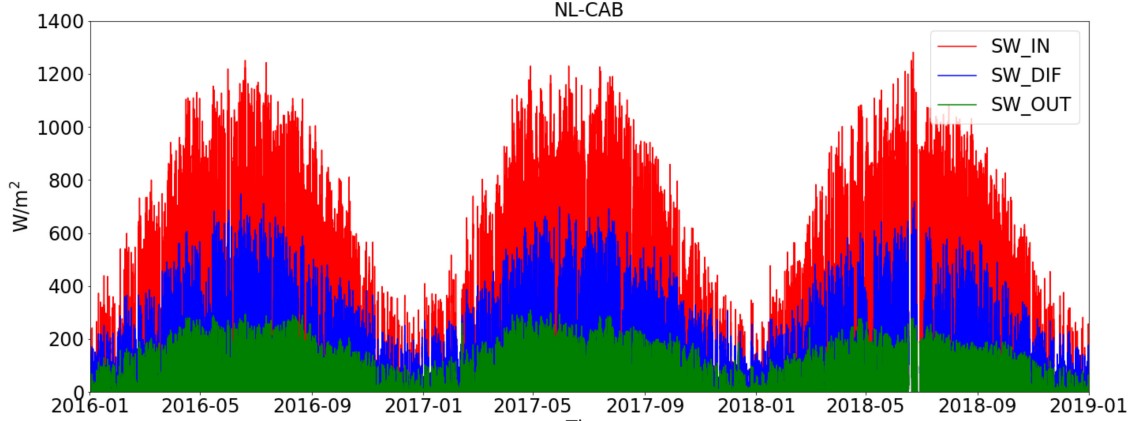

**Figure 4.** Time series of raw measurements at the NL-CAB tower site between 1 January, 2016 and 31 December, 2018. Incoming shortwave (SW_IN), outgoing shortwave (SW_OUT) and incoming diffuse shortwave radiation (SW_DIF) are represented by red, green and blue lines, respectively.

The raw data displayed in Figure 4 include a number of invalid measurements, e.g., radiation measurements with negative or missing values. The raw measurements need to be filtered before they can be used to calculate surface DHR and BHR. Figure 5 shows a comparison of reflectance variations between unfiltered and filtered albedometer measurements. This step filtered out all the SW_IN, SW_OUT and SW_DIF with radiation values smaller than 30 W/m$^2$. This threshold is obtained based on empirical experiments. In addition, measurements with a diffuse ratio (SW_DIF/SW_IN) out of the range [0, 1] are also filtered out. These filtered reflectances are then used to calculate surface DHR and BHR using the tower-based albedo estimation method introduced in [14]. This method uses a threshold of β (ratio of diffuse radiation to total incoming radiation) to extract only reflectances with a very small diffuse ratio to calculate black-sky-albedo (DHR, β<0.1), and only extract reflectances with a very large diffuse ratio in calculating white-sky-albedo (BHRs, β>0.9). This method has been demonstrated to work well over different landscapes, and is therefore not discussed further here [14].

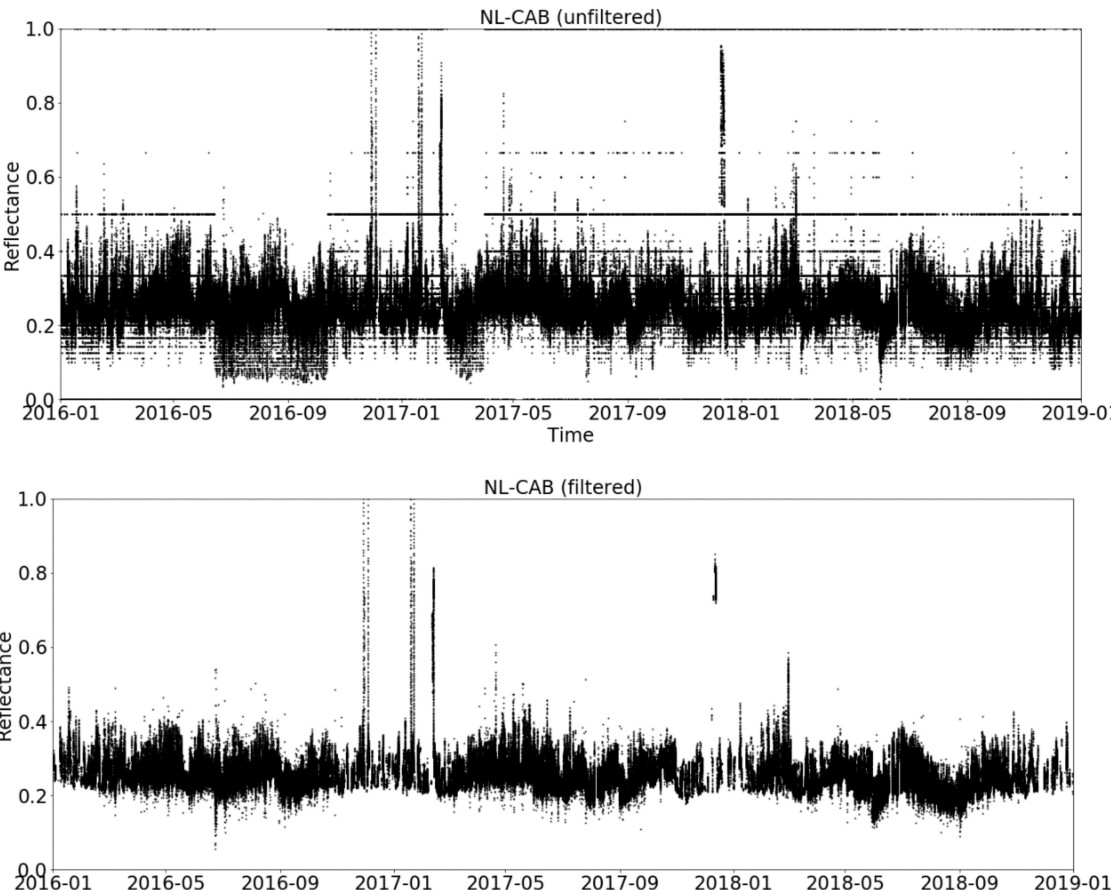

**Figure 5.** Comparison of unfiltered and filtered reflectance variations at the NL-CAB sites between the years 2016 and 2018.

## 2.2. Satellite Albedo

In this work, three different satellite albedo products are used to compare with tower-measured albedo values over the GbOV stations. They are the Copernicus Global Land Service (CGLS) albedo products at 1-km resolution, MODIS albedo products at 500 m resolution and MISR albedo products at 1.1 km resolution.

The CGLS albedo products are solely derived from the SPOT/VEGETATION instrument up until 2014, and since from the PROBA-V sensors [18]. The CGLS albedo products are updated every 10 days using a 30-day time window. The MODIS MCD43A3 products provide a collection of 500-m albedo data in the MODIS band 1-7, and the visible, near infrared and shortwave bands [3]. The MODIS 500-m albedo products are generated on a daily basis with a moving 16-day time window. The MISR instrument provides simultaneous albedo products that are retrieved from its multi-angular measurements [19]. The MISR albedo products are derived over four narrow spectral bands: blue (446 ± 21 nm), green (558 ± 15 nm), red (672 ± 11 nm) and near infrared (866 ± 20 nm). To obtain the broadband shortwave albedo, a narrow-to-broadband conversion using the following coefficients is needed:

$$\alpha^{MISR} = 0.126 \cdot \alpha_2 + 0.343 \cdot \alpha_3 + 0.415 \cdot \alpha_4 + 0.0037 \tag{1}$$

where $\alpha_2$, $\alpha_3$ and $\alpha_4$ represent MISR spectral albedos at band 2, 3 and 4, respectively where $\alpha^{MISR}$ is the derived broadband shortwave albedo [20].

### 2.3. The Sensor Invariant Atmospheric Correction (SIAC) method

SIAC, is a sensor agnostic approach to atmospheric correction, and was developed by Yin et al. [15]. It aims to provide consistent estimates of surface reflectance from different space-based high-resolution optical sensors. The basic idea of the SIAC method is to use the coarse-resolution spectral BRDF (500-m MODIS BRDF) to describe the surface anisotropy, and the Copernicus Atmospheric Monitoring Service (CAMS) 4.4 km data as a prior estimate of atmospheric composition. To retrieve the surface reflectance, an inverse problem needs to be solved using the following steps:

(1) MODIS MCD43A3 datasets provide 500-m, daily resolution spectral BRDF kernels, which can be used to derive the surface reflectance at the solar and viewing geometries that are consistent with the high-resolution satellite.

(2) As the spectral bands are different between the coarse-resolution BRDF (500-m MODIS) and high-resolution satellites (20-m Sentinel-2 or 60-m Landsat-8), a linear transformation is performed to convert the coarse-resolution surface reflectance to the target EO spectral bands.

(3) Due to the large differences in the spatial resolution between the MODIS and high-resolution EO, the surface reflectance from MODIS and top-of-atmosphere reflectance from high-resolution cannot be compared directly even when they are strongly correlated. Therefore, a point spread function (PSF) is modelled in order to make the coarse-resolution MODIS and high-resolution EO comparable.

(4) The coarse-resolution surface reflectance is mapped to the top of the atmosphere using a radiative transfer model, which can then be compared with the TOA reflectance convolved with the empirical PSF.

(5) An inverse problem is built to solve the aerosol optical thickness (AOT), total columnar water vapour (TCWV) and total columnar ozone (TCO3) based on a prior distribution from CAMS. This step also includes a spatial regularisation that smooths the spatial variation of atmospheric composition.

(6) The final step of SIAC is to correct the target TOA reflectance from high-resolution EO using the Lambertian surface-atmosphere coupling assumption and the atmospheric parameters inferred from above.

Details and validation results of the spectral transformation and spatial modelling have been presented in [15]. a flowchart in Figure 6 shows the inputs for SIAC and what processing is applied to each of these inputs.

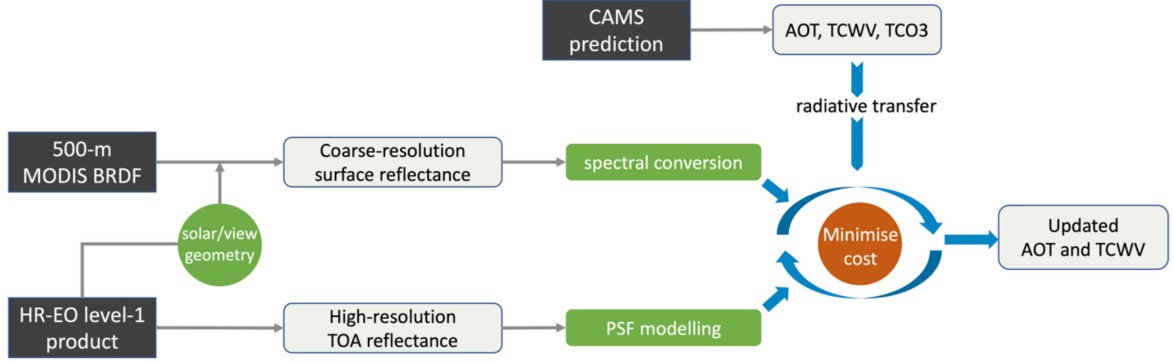

**Figure 6.** Flowchart of Sensor Invariant Atmospheric Correction (SIAC) processing chain to produce updated atmospheric state, which is subsequently employed to generate the surface spectral Bi-Directional Reflectance Factor (BRFs).

The main advantages that SIAC has over other atmospheric correction approaches are as follows: 1) it uses the MODIS BRDF as a physically meaningful constraint in retrieving a high-resolution

surface bi-directional reflectance; 2) it does not rely on AErosol RObotic NETwork (AERONET) aerosol measurements and uses CAMS data as a prior prediction. This is useful as not all the sites investigated in this study have AERONET measurements available; 3) the SIAC method produces an uncertainty estimation for every single high-resolution EO pixel. In this way, uncertainties caused by the atmospheric correction can be estimated properly when the surface albedo is upscaled from the tower to the coarse satellite resolution.

To illustrate the performance of SIAC atmospheric corrections, six 2D scatterplots are shown in Figure 7 to illustrate an intercomparison of Landsat-8 spectral surface reflectance derived from SIAC corrections and that derived from the LaSRC algorithm (https://github.com/USGS-EROS/espa-surface-reflectance/tree/master/lasrc). LaSRC is the specialized software that is used to produce Landsat-8 level-2 surface reflectance products. Surface reflectance derived from SIAC corrections display a good correlation with LaSRC corrections, except for a small bias at higher reflectance values. At low-reflectance values in bands 2, 3 and 4 (see Figure 7 caption for wavelengths), SIAC and LaSRC retrievals have larger uncertainties due to higher uncertainties arising from greater scattering.

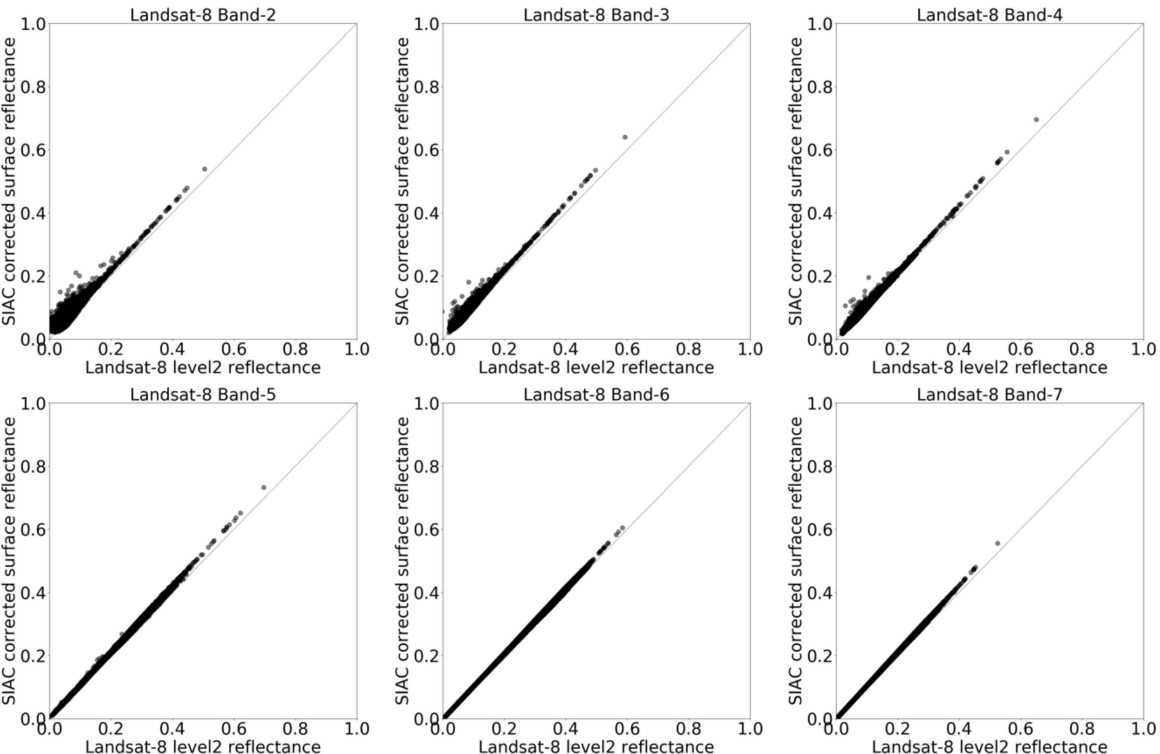

**Figure 7.** Intercomparison of Landsat-8 spectral surface reflectance derived from Landsat 8 Surface Reflectance Code (LaSRC) algorithm and SIAC processing at band 2 (482 nm), band 3 (561 nm), band 4 (655 nm), band 5 (865 nm), band 6 (1609 nm) and band 7 (2200 nm). This example uses the Landsat-8 scene in a 10 km * 10 km region that centres at the US-SXF site on 15th November 2018.

To evaluate the performance of SIAC corrections in retrieving Sentinel-2 surface reflectance values, three atmospheric correction models were used: SIAC, Sen2Cor and 6Sv. Sen2Cor (https://step.esa.int/main/third-party-plugins-2/sen2cor/) is a processor for Sentinel-2 level-2 product generation which includes retrieving atmospherically corrected surface reflectance. SIAC uses aerosol measurements based on CAMS prediction. Sen2Cor uses aerosol optical thickness derived at 550nm using the DDV (Dense Dark Vegetation) algorithm [21] based on the correlation between reflectance in the SWIR (band 12) and VIS (red - band 4, and blue - band 2). While comparing different atmospheric correction approaches, significant discrepancies were found between SIAC and Sen2Cor results. Therefore, the 6Sv model which uses near-real-time Aerosol Optical Depth (AOD) measurement from AERONET in atmospheric corrections is employed in this example as a way to provide reference values of surface

reflectance. The intercomparison of surface reflectance derived from the three approaches is shown in Figure 8. It is clearly seen that the SIAC corrections have closer agreement with reference values (6Sv retrievals) than the Sen2Cor corrections.

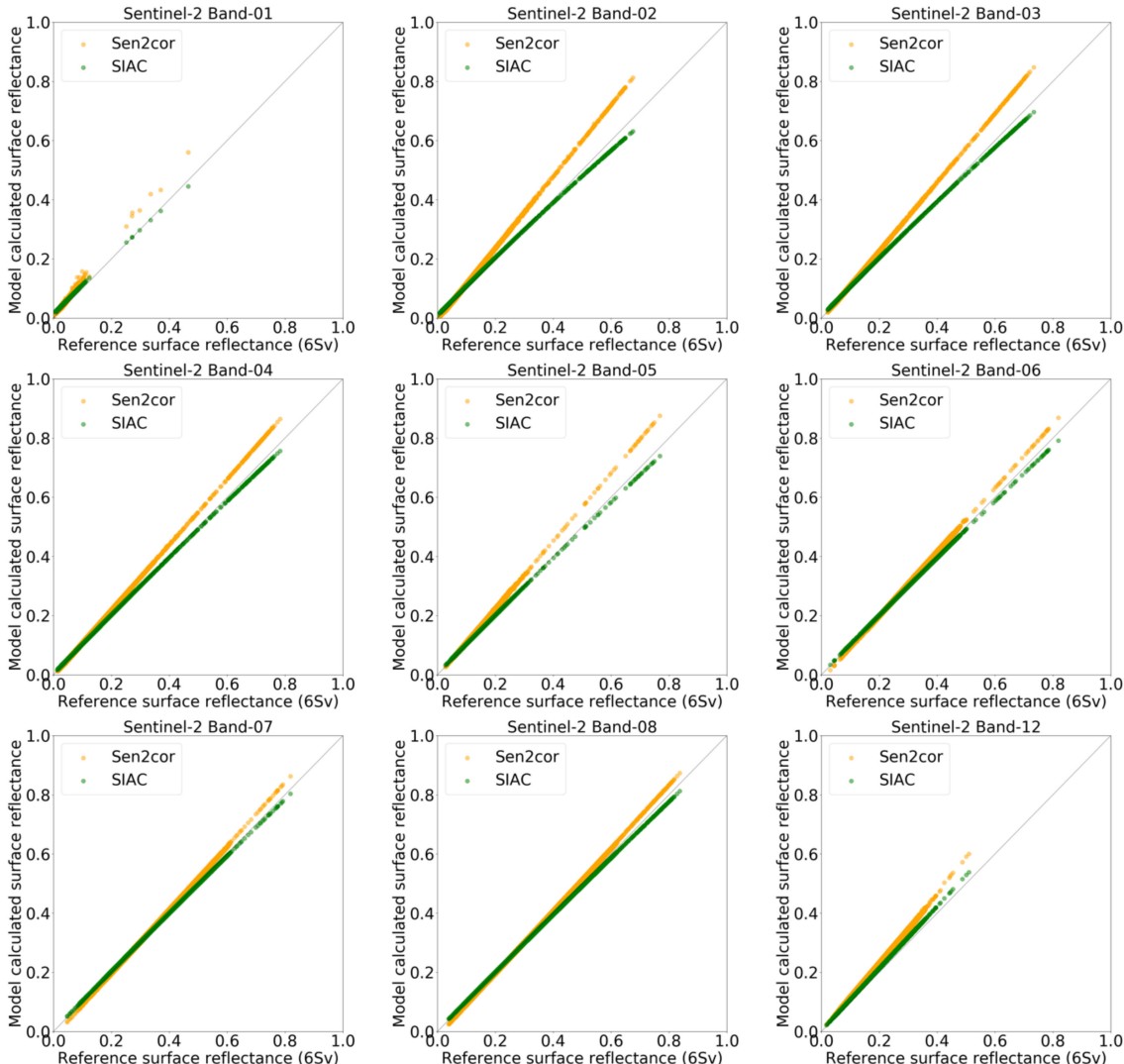

**Figure 8.** Intercomparison of Sentinel-2 spectral surface reflectance derived from 6Sv, Sen2Cor and SIAC atmospheric corrections at band 1 (442.7 nm), band 2 (492.4 nm), band 3 (559.8 nm), band 4 (664.6 nm), band 5 (704.1 nm) and band 6 (7 40.5 nm), band 7 (782.8 nm), band 8 (832.8 nm) and band12 (2202.4 nm). This example uses the Sentinel-2A scene in a 10 km * 10 km region that centres at the US-SXF site on 2nd July 2017.

In Figure 9, an example of Sentinel-2 level-1 TOA reflectance and level-2 Bottom Of Atmosphere (BOA) reflectance derived from SIAC is shown.

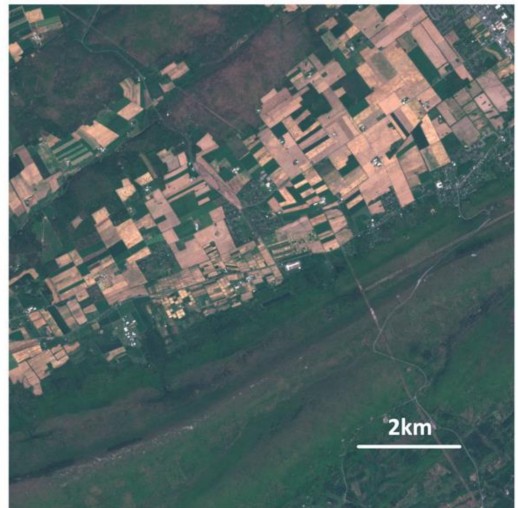 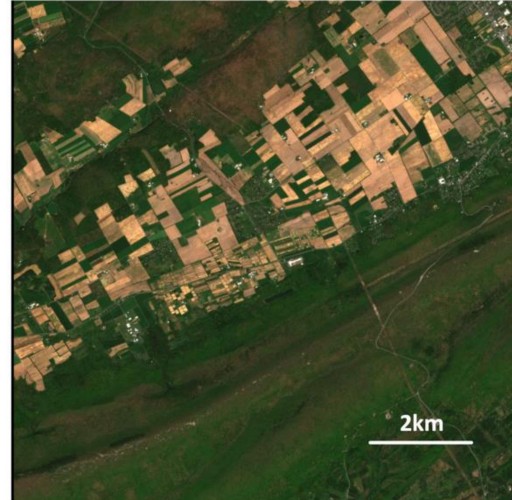

RGB composition of TOA reflectance          RGB composition of BOA reflectance

**Figure 9.** Example of Sentinel-2A level1 Top Of Atmosphere (TOA) reflectance (RGB composition) and BOA reflectance (RGB green-red-blue composition) derived from SIAC corrections in a 10 km * 10 km region that centres at the US-PSU site on 9 May, 2018.

## 2.4. Uncertainty Estimation

To infer the uncertainties in surface albedo upscaled from in situ retrievals to coarse satellite resolutions, there are two main uncertainty sources that need to be considered. Firstly, in situ albedo values are retrieved based on albedometer measured raw measurements, which are downwelling, upwelling and diffuse shortwave radiation at a time frequency ranging from 1 minute to 1 hour. The uncertainty in raw radiation measurements is defined by the albedometer specification. For albedometers at tower sites investigated in this study, typically they have a daily relative uncertainty ($\sigma_{daily}$) of 5% at the 95% confidence level. As the in situ albedo generation combines effective raw measurements over a fixed time window (e.g. 30 days for CGLS products), the relative uncertainty in the produced albedo can be reduced to:

$$\sigma_{\text{tower\_albedo}} = \frac{\sqrt{2} \times \sigma_{\text{daily}}}{\sqrt{N}} \tag{2}$$

where N is the number of effective days in albedo calculation, and the multiplication factor 2 indicates contributions from both downwelling and upwelling shortwave radiation.

The second major uncertainty comes from the high-resolution EO, which is used as a bridge to fill gaps between the small footprint tower measurements and the coarser-resolution satellite measurements [14]. The high-resolution spectral surface reflectance derived from SIAC corrections comes with uncertainty estimations for every single pixel. This uncertainty takes into account the effect of the atmosphere in transforming albedo values from in situ retrievals to coarse EO scales. However, narrow-to-broadband conversion coefficients [20,22] are needed to convert uncertainties in spectral reflectance to shortwave reflectance. Equation (3) and (4) list the calculation of shortwave broadband uncertainties for Landsat-8 and Sentinel-2, respectively.

$$\sigma_{\text{L8\_SW(absolute)}} = \sqrt{0.356^2\sigma_{\text{L(b2)}}{}^2 + 0.13^2\sigma_{\text{L(b4)}}{}^2 + 0.373^2\sigma_{\text{L(b5)}}{}^2 + 0.085^2\sigma_{\text{L(b6)}}{}^2 + 0.072^2\sigma_{\text{L(b7)}}{}^2} \tag{3}$$

$$\sigma_{\text{S2\_SW(absolute)}} = \sqrt{0.356^2\sigma_{\text{S(b2)}}{}^2 + 0.13^2\sigma_{\text{S(b4)}}{}^2 + 0.373^2\sigma_{\text{S(b8)}}{}^2 + 0.085^2\sigma_{\text{S(b11)}}{}^2 + 0.072^2\sigma_{\text{S(b12)}}{}^2}$$
$$\tag{4}$$

where in Equation (2) $\sigma_{L(b2)}$, $\sigma_{L(b4)}$, $\sigma_{L(b5)}$, $\sigma_{L(b6)}$ and $\sigma_{L(b7)}$ represent the Landsat-8 pixel-level surface reflectance uncertainty in bands 2, 4, 5, 6 and 7, respectively. Similarly, in Equation (3), $\sigma_{S(b2)}$, $\sigma_{S(b4)}$, $\sigma_{S(b8)}$, $\sigma_{S(b11)}$ and $\sigma_{S(b12)}$ represent the Sentinel-2 pixel-level surface reflectance uncertainty in bands 2, 4, 8, 11 and 12, respectively. The calculated broadband reflectance uncertainty is the absolute values, which should be converted to relative values following Equation (5) such that uncertainties from the albedometer and high-resolution EOs can be compared.

$$\sigma_{\text{HREO\_SW}} = \sigma_{\text{SW(absolute)}} / \text{BRF}_{\text{SW}} \tag{5}$$

where $\sigma_{HREO\_SW}$ is the relative uncertainty of shortwave surface albedo from high-resolution EO observations. It is derived from dividing its absolute uncertainty by the corresponding shortwave broadband reflectance.

Finally, the uncertainty in upscaled coarser-resolution albedo can be calculated by combining the contribution from tower albedometer and high-resolution EO measurements as follows:

$$\sigma = \sigma_{\text{tower\_albedo}} \tag{6}$$

Figure 10 shows DHRs that are upscaled to the CGLS 1km * 1km resolution between year 2017 and 2018 with estimated uncertainty values. The tower-based albedo retrievals are upscaled at selected dates when cloud-free high-resolution EOs are available. Similarly, the upscaled BHRs and corresponding uncertainties are displayed in Figure 11.

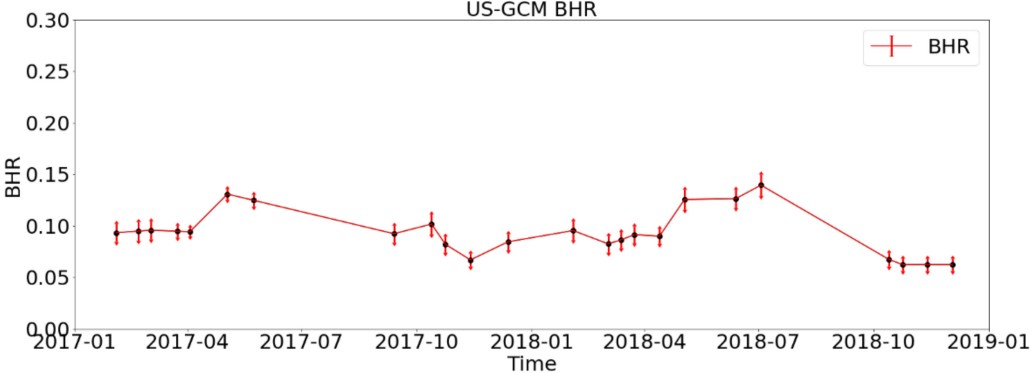

**Figure 10.** Example of upscaled Direct Hemispherical Reflectance (DHR) and uncertainty values at the US-GCM site.

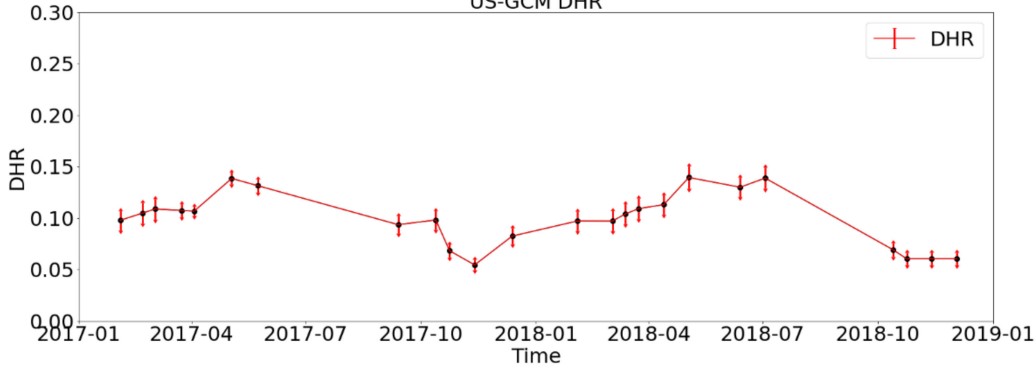

**Figure 11.** Example of upscaled bi-hemispherical reflectance (BHRs) and uncertainty values at the US-GCM site.

## 3. Results

*3.1. Comparison of Surface Albedo between Satellite Products, In Situ Retrievals and Upscaled Values*

Song et al. [14] presented the comparison of surface albedo between satellite product and in situ retrievals mainly at SURFRAD and BSRN sites, with only one FLUXNET site at AU-TUM (see Table 1). In this study, albedo comparisons at all available sites in Table 1 are presented. Site-specific in situ DHR and BHR network retrievals use the same method derscribed in [14]. The intercomparison between in situ tower retrievals and coarse-resolution satellite products are presented in Figure 12 for DHRs and Figure 13 for BHRs. This intercompairson includes three sites, which are DE-HAI from the FLUXNET network where the surface type is mixed forest, NL-CAB from the BSRN network where the surface type is grassland, and US-FPK from the SURFRAD network where the surface type is grassland as well. Intercomparisons between all the other tower and satellite-measured albedos are presented in the Supplementary Materials.

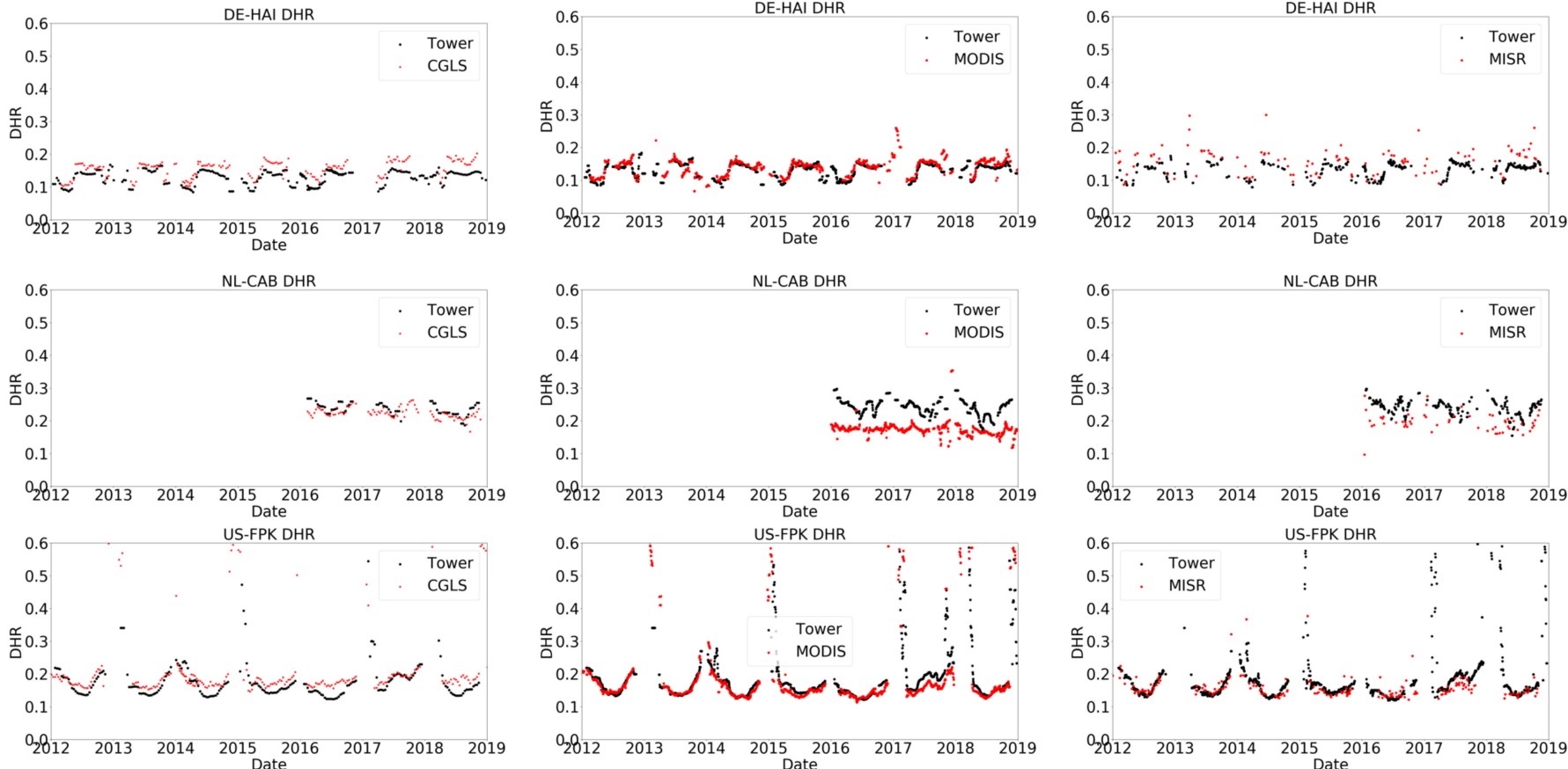

**Figure 12.** Intercomparison of DHR values derived from in situ tower measurements and coarse-resolution satellite products at the DE-HAI, NL-CAB and US-FPK sites. Copernicus Global Land Service (CGLS) values are depicted in the first column, moderate resolution imaging spectroradiometer (MODIS) values are depicted in the second column and multi-angle imaging spectroradiometer (MISR) values are depicted in the third column.

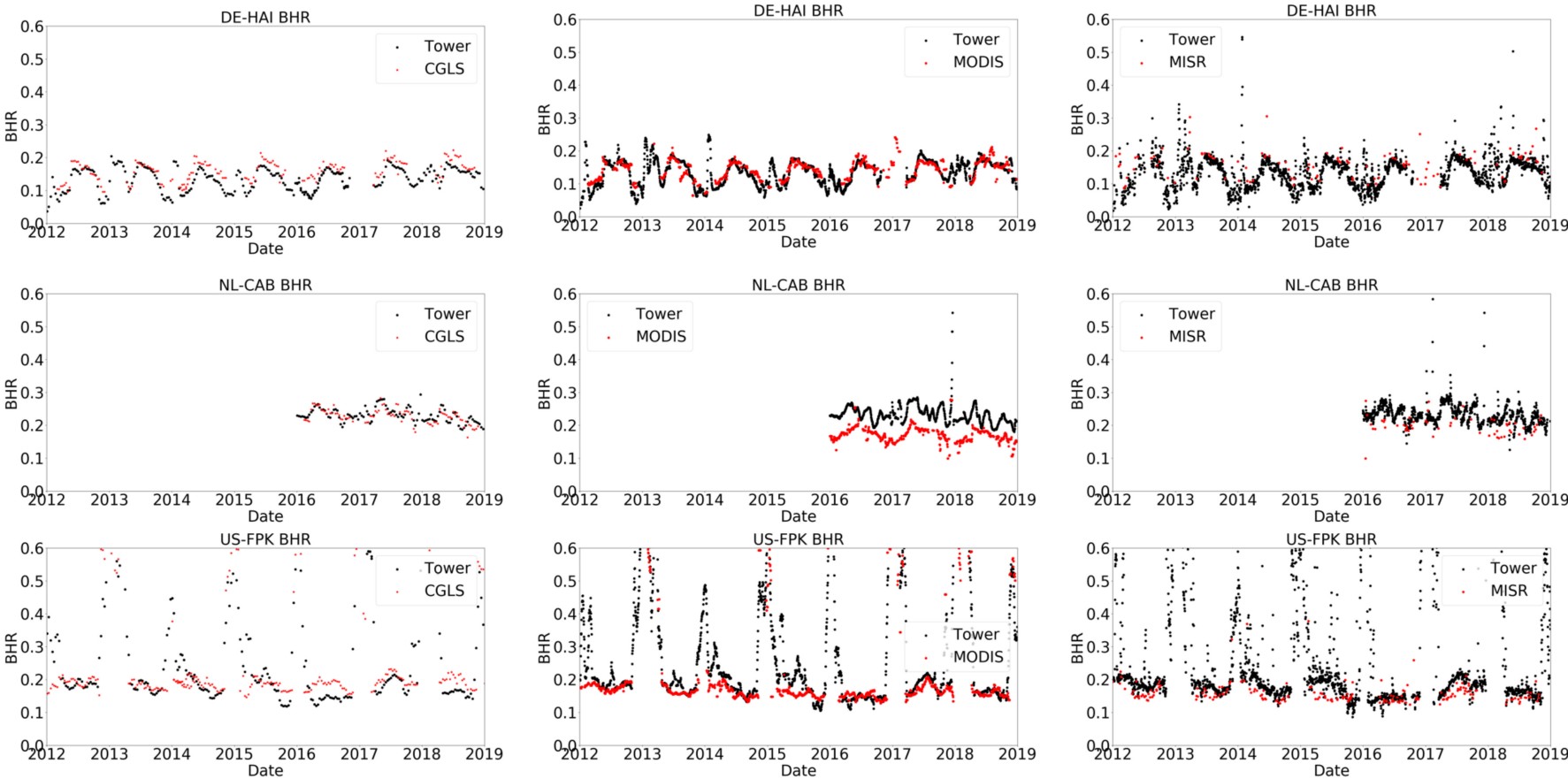

**Figure 13.** Intercomparison of BHR values derived from in situ tower measurements and coarse-resolution satellite products at the DE-HAI, NL-CAB and US-FPK sites. CGLS values are depicted in the first column, MODIS values are depicted in the second column and MISR values are depicted in the third column.



Tower-measured albedos were produced over three different time windows corresponding to the integration period employed to generate the satellite measurements, with the following periods: 16 days for the comparison with daily MODIS albedos; 30 days every 10 days for the comparison with CGLS albedos; and near simultaneously for the comparison with the 7 minute MISR albedos. Generally speaking, the tower-measured albedos show good agreement with the satellite-measured albedos throughout the 7 years (2012–2018). At the At the DE-HAI forest site, MODIS DHRs and BHRs agree with tower-measured DHRs and BHRs in terms of absolute values and seasonal variations. This results in a R-squared ($R^2$) value of 0.65 between tower measured and MODIS retrieved BHRs at DE-HAI. CGLS DHRs have a good match with tower-measured albedos between year 2012 and 2014, but are slightly overestimated between year 2015 and 2018. This may be related to the switchover from VEGETATION to Proba-V in 2014. The R-squared ($R^2$) value between tower measured and CGLS retrieved BHRs is 0.68 at DE-HAI. MISR retrievals also show overestimated DHRs throughout the period. At the US-FPK grassland, MODIS and MISR DHRs/BHRs are consistent with tower-measured DHRs/BHRs in the snow-free season, while CGLS DGRs/BHRs are still overestimated. This results in a R-squared ($R^2$) value of 0.88 in MODIS retrievals and a R-squared ($R^2$) value of 0.72 in CGLS retrievals at US-FPK. In the snow-covered season, MODIS retrievals have better performance than CGLS retrievals in picking up the snow, probably due to the daily sampling. An exception is the NL-CAB grassland where there is a large difference between MODIS retrieved DHRs/BHRs and tower-measured DHRs/BHRs. This is likely to do with an issue in MODIS BRDF modelling or residual cloud effects over that area.

By comparing tower satellite measured albedos at the GbOV tower sites investigated in this study, MODIS retrievals appear to have the best agreement with tower retrievals amongst the three, except at the NL-CAB site. At the NL-CAB site, the MODIS values appear lower than the tower values because this area has persistent cloudiness at the MODIS overpass times. The DHRs have a better match between tower and satellite retrievals than BHRs, because tower DHRs are calculated over a narrow time-window around local solar noon, whereas BHRs are calculated using measurements acquired throughout the day. Moreover, the tower-based BHRs are calculated based on the assumption that illumination from the sky is uniform from all angles when the diffuse ratio is very large ($\approx 0.9$). However, this uniform illumination condition is hard to meet with real observations. This explains why tower-measured DHRs appear to have a better agreement with satellite measurements than the BHRs.

Intercomparisons for other GbOV tower sites are included in the Supplementary Materials, except for the DOM and NO-NYA as they are outside the 75°N and 60°S boundary of the CGLS observations. We can see that the DHRs at the AU-TUM show a good agreement between tower and MODIS retrievals from year 2012 to 2016, whereas CGLS retrievals are strongly overestimated. CGLS also has overestimated BHR retrievals at the US-DRA while MODIS retrievals are consistent with tower-measured values.

## 3.2. Comparison of Surface Albedo between Satellite Products and Upscaled Values

In [14], intercomparisons of DHRs and BHRs between upscaled tower retrievals and the three satellite retrievals are already presented at a subset of the individual tower sites, and therefore are not discussed further here. Statistical results showing a comparison of BHRs upscaled from the tower using the SIAC atmospheric correction and coarse-resolution CGLS albedo products are displayed in Figure 14 at selected GbOV tower sites with different land covers, including closed shrublands, evergreen broadleaf, mixed forest, croplands, grasslands, bare soil and rocks. The upscaled albedo has the same spatial scale as the coarse resolution satellite retrievals, so that they can be compared at the per pixel level directly. Statistical analysis of upscaling tower-measured albedo to coarse resolutions over a larger area around the tower albedometer is shown in Figures 11 and 12 from [14].

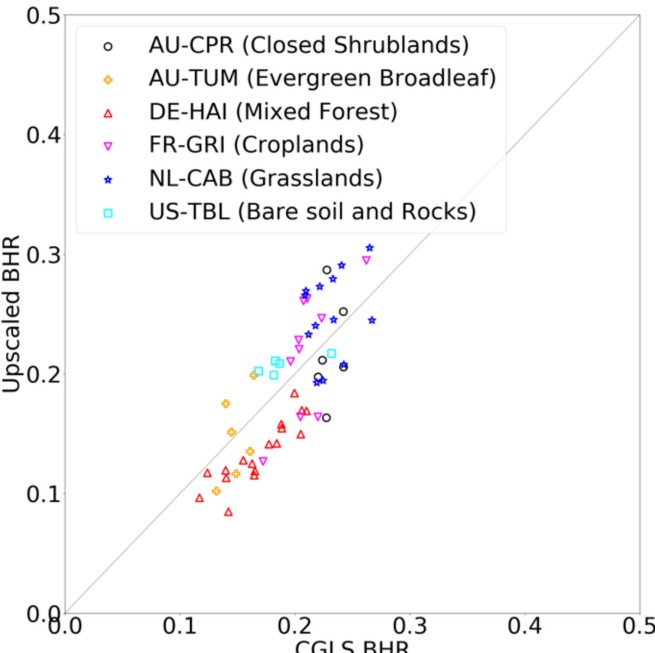

**Figure 14.** Summary of albedo results for BHRs upscaled from tower to CGLS spatial scale and BHRs from CGLS retrievals at selected tower stations with different land covers.

## 4. Discussion

The DHR and BHR retrievals which were produced at 20 GbOV tower stations between 2012 and 2016 [14], are extended to 2018. The original DE-GEB, US-BAO, US-BRW and SPO sites are replaced by the NL-CAB, NM-GOB, NO-NYA and DOM in the new time period. The GbOV tower stations cover a wide range of different land cover types including forests, croplands, grasslands, desert etc. DHRs and BHRs derived from tower albedometer measurements at these stations were compared with satellite retrievals including CGLS, MODIS and MISR. Among the three satellite retrievals, MODIS has the best agreement with tower measurements. MODIS retrievals can pick up some snow, which is most likely due to its daily sampling. MISR retrievals can occasionally pick up snow which can be seen in the US-FPK and US-SXF. The CGLS albedo retrievals are affected by the switchover from VEGETATION to Proba-V in 2014, and are best displayed from the US-DRA. DHR retrievals often have better agreement between tower and satellites than BHR retrievals for two reasons: 1). DHRs are measured at a specific time of day while BHRs are measured throughout the day; 2). The idealised uniform illumination condition is hard to meet when measuring BHRs.

There are several sites which show anomalous behaviour. For example, the lower MODIS values at the NL-CAB site due to persistent cloudiness at the MODIS overpass times. The supplementary plots of tower DHR vs EO show that all the EO sometimes indicates a positive bias (BE-BRA, IT-REN) and more often a negative bias (FR-GRI, AU-CPR, US-BON, US-PSU, US-CGM) some of which is also reflected in the BHR plots. Snow is not well captured by CGLS and this probably relates to the 30-day integration time period.

In [14] an assessment was shown of DHR and BHR from a larger area centred on one site (US-SXF) in order to produce sufficient statistics. Here, a summary plot is shown of results from different land covers, which indicate that overall there is good agreement between the CGLS and upscaled albedo values. However, there are insufficient data points to produce a quantitative set of statistics.

A combination of coarse-resolution MODIS BRDF climatology and HR-EO surface reflectance was used to retrieve HR surface albedos, which was subsequently used to upscale tower albedos. The HR-EO surface reflectances were produced using the SIAC atmospheric correction method, which includes a representation of anisotropy or surface directional/structural/topography dependence.

The SIAC method was demonstrated to have a good agreement with the LaSRC algorithm in retrieving Landsat-8 spectral surface reflectance, and better than the Sen2cor tool in retrieving Sentinel-2 spectral surface reflectance. The SIAC method also produces an uncertainty estimation of retrieved spectral surface reflectance at the HR-EO pixel level. The SIAC derived surface reflectance uncertainty along with the albedometer radiation measurement uncertainty, are combined to provide an estimate of upscaled albedo uncertainty in coarse resolutions.

Figure 3 in the Supplementary Materials shows the tower FoV, CGLS 1km grid and MODIS 1km grid at individual tower sites. The albedo upscaling process uses HR-EO albedo retrievals to fill the spatial gap between the tower FoV and coarse-resolution satellite observations. Figure 14 shows the comparison of surface BHRs between upscaled tower measurements and CGLS retrievals over both homogeneous and heterogeneous. This method provides direct way for comparing surface albedo retrieved from different platforms with different spatial scales.

## 5. Conclusions

In this study, the method developed by Song et al. [14] for upscaling surface albedo from tower albedometer FoV to coarse satellite resolutions is further refined and demonstrated for heterogeneous sites in addition to homogenous sites. The new upscaling method uses high-resolution EO surface reflectance measurements (Landsat-8 and Sentinel-2) as a resolution bridge to fill the gaps between tower albedometer FoV and coarse satellite spatial scales, where high-resolution EO surface reflectances are retrieved using the novel SIAC atmospheric correction approach. The tower derived albedos as well as the upscaled albedos, are compared against three satellite observations (CGLS, MODIS and MISR) at 20 GbOV sites between year 2016 and 2018. Uncertainties in upscaled albedos are estimated by considering uncertainties from both the tower albedometer raw measurements and SIAC atmospheric corrections. The upscaled albedo has the largest absolute uncertainty of about 0.02.

The GbOV tower sites presented in this study have different types of land covers, and include both homogeneous and heterogeneous landscapes. Experimental results demonstrate that the method proposed in this paper can provide a way to validate satellite albedo retrievals using tower albedometer measurements over both homogeneous and heterogeneous landscapes. This overcomes the previous limitation, such as those recommended by the CEOS-LPV (Land Product Validation) protocol that satellite albedo retrievals can only be validated by in situ measurements only over homogeneous surfaces. In addition, we demonstrate that the uncertainties of upscaled albedos can be traced from the in situ instrument through to the upscaled satellite measurements through the SIAC atmospheric correction providing uncertainties of retrieved surface reflectances at the per pixel level. This paper also demonstrates the method to generate high-resolution surface albedo based on 1-km MODIS BRDF climatology and SIAC derived high-resolution spectral surface reflectances. In future work, we plan to study how accurate high-resolution (≤20 m) albedo can be retrieved based on this MODIS BRDF and high-resolution EO combination for Sentinel-2 MSI.

**Supplementary Materials:** The following are available online at http://www.mdpi.com/2072-4292/12/5/833/s1.

**Author Contributions:** Conceptualization, J.-P.M.; Methodology, J.-P.M. and R.S.; Software, R.S.; Validation, J.-P.M. and R.S.; Formal Analysis, R.S.; Investigation, R.S.; Resources, J.-P.M.; Data Curation, J.-P.M., R.S. and S.K.; Writing-Original Draft Preparation, R.S.; Writing-Review & Editing, J.-P.M., S.K., F.Y.; W.W., M.K., M.R., N.A., N.G., W.M., G.K., D.B., B.B., A.K., L.S., P.B., B.L., M.L., C.L. Visualization, R.S.; Project Administration, J.-P.M.; Funding Acquisition, J.-P.M. All authors have read and agreed to the published version of the manuscript.

**Funding:** This work was funded by the European Commission Joint Research Centre contract FWC932059, part of the Global Component of the European Union's Copernicus Land Monitoring Service. This work has been undertaken using data processed by the team at UCL for the GBOV project "Ground Based Observation for Validation" (https://land.copernicus.eu/global/gbov). Funding for the AmeriFlux core site US-NR1 data was provided by the U.S. Department of Energy's Office of Science.

**Acknowledgments:** For processing of satellite data, we thank JASMIN, the UK's collaborative data analysis environment http://jasmin.ac.uk. We would like to thank Christian Lanconelli of JRC Ispra for fruitful discussions. We would like to thank NOAA for access to their datasets through SURFRAD (http://www.esrl.noaa.gov/gmd/grad/surfrad/) and the BSRN (http://bsrn.awi.de) for access to their datasets. This work also used tower albedometer

data acquired by the FLUXNET community and in particular by the following networks: AmeriFlux (U.S. Department of Energy, Biological and Environmental Research, Terrestrial Carbon Program (DE-FG02-04ER63917 and DE-FG02-04ER63911), CarboEuropeIP, CarboItaly, CarboMont, OzFlux, USCCC. GBOV product developments are managed by ACRI-ST from the research work of University College London, University of Leicester, University of Southampton, University of Valencia and Informus GmbH. We acknowledge the financial support to the tower albedometer data harmonization provided by CarboEuropeIP, FAO-GTOS-TCO, iLEAPS, Max Planck Institute for Biogeochemistry, National Science Foundation, University of Tuscia, Universiteé Laval and Environment Canada and US Department of Energy and the database development and technical support from Berkeley Water Center, Lawrence Berkeley National Laboratory, Microsoft Research eScience, Oak Ridge National Laboratory, University of California-Berkeley, University of Virginia. Nicola Arriga acknowledges support and funding received from ICOS-Belgium.

This work used data acquired as part of the principal eddy covariance data which is shared by the FLUXNET community, including these networks: AmeriFlux, OzFlux-TERN. The FLUXNET eddy covariance data processing and harmonization was carried out by the European Fluxes Database Cluster, AmeriFlux Management Project, and Fluxdata project of FLUXNET, with the support of CDIAC and ICOS Ecosystem Thematic Center, and the OzFlux offices.

The MODIS data should be referenced via DOI: 10.5067/MODIS/MCD43A3.006. See the Supplementary Materials for the web location to download all the input datasets from.

**Conflicts of Interest:** There are no conflicts of interest.

## Abbreviations

The following abbreviations are used in this manuscript:

| | |
|---|---|
| Aerosol Optical Depth | AOD |
| AErosol RObotic NETwork | AERONET |
| Baseline Surface Radiation Network | BSRN |
| Bi-Hemispherical Reflectance | BHR |
| Bi-Directional Reflectance Factor | BRF |
| Bidirectional Reflectance Distribution Function | BRDF |
| Bottom Of Atmosphere | BOA |
| Copernicus Global Land Service | CGLS |
| Dense Dark Vegetation | DDV |
| Directional Hemispherical Reflectance | DHR |
| Earth Observing System | EOS |
| Field-of-View | FoV |
| Ground Based Observation for Validation | GbOV |
| International Geosphere-Biosphere Programme | IGBP |
| Landsat 8 Surface Reflectance Code | LaSRC |
| Moderate Resolution Imaging Spectroradiometer | MODIS |
| Multi-Angle Imaging Spectroradiometer | MISR |
| Point Spread Function | PSF |
| Surface Radiation Budget Network | SURFRAD |
| Top Of Atmosphere | TOA |
| World Climate Research Programme | WCRP |
| World Meteorological Organization | WMO |

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
