# Peer review of "Validation of Space-Based Albedo Products from Upscaled Tower-Based Measurements Over Heterogeneous and Homogeneous Landscapes"

_remotesensing, doi:10.3390/rs12050833_

Round 1

Reviewer 1 Report

This paper is very well written and its subject is significant.  There are many acronyms, but it is still not too difficult to read.  Several suggestions for improvement:

The error analysis in section 2.4 deserves more discussion.  In particular, are daily errors independent as implied in Eq. (2)? What about Eq. (6)? Some of the figures are nearly impossible to read as they are presently presented (e.g., Figs. 12, 13).  Consider alternate layouts or publication in landscape.  Fig. 14 is marginally legible.

Reviewer 2 Report

This paper gives a comprehensive assessment of tower-based albedo products over various land types and heterogeneity -- upscaled to satellite pixel resolution. These are validated through their comparison to existing satellite-based albedo products.  This work is important for understanding the extent to which ground-based albedo reference measurements can be extended to larger spatial scales. Most of the comments below relate to issues with the uncertainty analysis, and clarifications needed on the methodology and results. Please revise accordingly or otherwise address:

Main:

2.4: There appear to be errors/omissions in your uncertainty analysis: Equation (2) should have a factor of 2^1/2, not 2. How are the uncertainties of the SAIC-based spectral surface reflectance derived? The variables in Equations 3 and 4 are the narrow band albedos of the various bands, not their uncertainties. To determine the combined uncertainty, you would use the propagation of uncertainty formula. Likewise, in equation 5, the values should be root sum squared, not just summed. 2.4: Can you be more specific about how the high resolution EO is used to fill gaps in the tower measurement? General: Please include a discussion on whether the differences in the various comparisons are compatible with the estimated uncertainties (Are the uncertainties of the comparisons less than their differences?).

Others:

2.1: Why is 30 W/m^2 chosen as the threshold value. Is this the noise level of the instrument? 2.3: Please include more description on the PSF model used to make the high and lower resolution images comparable. For instance, how do you derive/estimate the MODIS PSF to model the high resolution scene as detected by MODIS? Is it the method used in Ref. 15? 2.3: “SIAC corrections have strong correlations with LaSRC correction …” You could point out that there is also a bias between them and quantify it. 2.3: Are you suggesting that the Sen2Cor has higher uncertainty because you must rely on its own aerosol inputs as opposed to 6sV where you can input the measured aerosol values? Perhaps you could clarify this point by describing the aerosol inputs used for the Sen2Cor model. 3.1: For the MODIS-CGLS comparison, I would recommend quantifying the agreement instead of just labeling it “good agreement.” 3.1: It would probably be clearer to explain the larger MODIS uncertainty here instead of in the discussion section (persistent cloudiness). 3,2: The second half of the first paragraph is repeated from earlier. Discussion: You noted that good agreement is shown but quantitative assessment can not be made. Perhaps it could by combining the data for all BHR values. For instance, you could quote the average percent difference +/- the standard deviation of the mean. Abstract: You claim that the upscaling to coarser resolution is a new method. More details on the method including PSF estimation are needed to support this statement. If this is indeed the same method as used in Ref. 15, you would need to qualify this statement. (Potential alternative statement: “We implemented a (or our) recently developed method.. [15]…”) Abstract: “novel atmospheric correction.” It seems like more of an improvement/refinement of an existing method.

Round 2

Reviewer 2 Report

Thank you for your effort in revising the paper. I believe it has been greatly improved. I would still like to see a few points clarified/corrections made pertaining to the following:

I still believe there is an error in Equation 2. Relative uncertainties are not additive (unless they are biases). If you are dividing two values – the upwelling by the downwelling radiance in this case – you can use the standard propagation of uncertainty formula and divide the entire equation by the albedo squared to convert to relative uncertainties: ualb/alb^2 = uLdown^2/Ldown^2 + uLup^2/Lup^2 (where alb is albedo, Ldown is downwelling radiance and Lup is upwelling radiance). If the relative uncertainties u/L are equal, you get a ualb/alb = 2^1/2*u/L. So there still should be a factor of 2^1/2, not 2.

I am still not clear why you claim that there is no bias shown in Figue 7. The Figure shows the SAIC correction above the 1-1 line, especially at higher reflectance values. To me, the differences at lower reflectance values are more an indication of  higher uncertainty (larger scatter).  I still suggest some clarification here.
